# Characterization of Two NMN Deamidase Mutants as Possible Probes for an NMN Biosensor

**DOI:** 10.3390/ijms22126334

**Published:** 2021-06-13

**Authors:** Alessandra Camarca, Gabriele Minazzato, Angela Pennacchio, Alessandro Capo, Adolfo Amici, Sabato D’Auria, Nadia Raffaelli

**Affiliations:** 1Institute of Food Sciences, National Research Council, Via Roma 64, 83100 Avellino, Italy; alessandra.camarca@isa.cnr.it (A.C.); angela.pennacchio@isa.cnr.it (A.P.); alessandro.capo@isa.cnr.it (A.C.); 2Department of Agricultural, Food and Environmental Sciences, Polytechnic University of Marche, Via Brecce Bianche, 60131 Ancona, Italy; g.minazzato@pm.univpm.it; 3Department of Clinical Sciences DISCO, Section of Biochemistry, Polytechnic University of Marche, Via Brecce Bianche, 60131 Ancona, Italy; a.amici@staff.univpm.it; 4Department of Biology, Agriculture and Food Science, CNR, Piazzale Aldo Moro 7, 00125 Rome, Italy

**Keywords:** nicotinamide mononucleotide, PncC, NMN deamidase, molecular recognition element

## Abstract

Nicotinamide mononucleotide (NMN) is a key intermediate in the nicotinamide adenine dinucleotide (NAD+) biosynthesis. Its supplementation has demonstrated beneficial effects on several diseases. The aim of this study was to characterize NMN deamidase (PncC) inactive mutants to use as possible molecular recognition elements (MREs) for an NMN-specific biosensor. Thermal stability assays and steady-state fluorescence spectroscopy measurements were used to study the binding of NMN and related metabolites (NaMN, Na, Nam, NR, NAD, NADP, and NaAD) to the PncC mutated variants. In particular, the S29A PncC and K61Q PncC variant forms were selected since they still preserve the ability to bind NMN in the micromolar range, but they are not able to catalyze the enzymatic reaction. While S29A PncC shows a similar affinity also for NaMN (the product of the PncC catalyzed reaction), K61Q PncC does not interact significantly with it. Thus, PncC K61Q mutant seems to be a promising candidate to use as specific probe for an NMN biosensor.

## 1. Introduction

The mononucleotide form of nicotinamide, i.e., nicotinamide mononucleotide (NMN), is a central metabolite in NAD biosynthesis. In particular, the NAD biosynthetic routes starting from nicotinamide (Nam) and nicotinamide riboside (NR) converge to NMN, which then undergoes adenylation to NAD. Several studies have demonstrated that NMN supplementation enhances NAD biosynthesis in various tissues under normal and pathological conditions [1]. As a consequence, in preclinical studies, NMN supplementation has shown to have beneficial effects on a large array of diseases, including metabolic, vascular, and neurodegenerative disorders [2]. However, our knowledge is very poor on NMN pharmacokinetics. The cellular NMN uptake is still a matter of investigation, and some studies have shown that NMN needs to be processed to NR prior to entering the cell [3,4], whereas other studies support its direct uptake through a dedicated transporter [5,6]. Likewise, only limited data, mostly from metabolic flux studies, are available on NMN effectiveness in different tissues. The complexity of the quantification of NMN has not yet allowed one to develop a clear picture of its cellular availability. Its presence in plasma remains controversial, as NMN is reported to be undetectable by some authors or to range from 7 nM to about 50 µM by others [3,7,8]. In human cerebrospinal fluid and in ascites exudates of tumor-bearing mice, nanomolar concentrations of NMN have been determined [9,10]. In tissues, NMN levels are much lower than NAD levels, which generally range from 0.1 to 1 mM. In particular, NMN/NAD ratios from 1:120 to 1:3000 have been determined, depending on the type of tissue [11]. Concentrations of 1.3 and 33 µM NMN have been measured in a glioma cell line [12] and in mouse erythrocytes [13], respectively. It is now becoming evident that the intracellular concentration of NMN should be finely controlled. In fact, in neurons, the increase of the NMN/NAD ratio, due to an increase of NMN or a decrease of NAD, can promote the activation of the pro-degenerative protein SARM1, thus triggering axon destruction [14,15].

Currently, NMN quantification is mostly performed by fluorometric or HPLC/MS techniques. Several HPLC–UV methods, already exist to measure NAD metabolites levels with a LOQ ranging from mM to µM [16]. Analytical methods based on liquid chromatography–tandem mass spectrometry [13], SRM–LC MS [17] or RP–UHPLC–MS/MS [18] are suitable for simultaneous detection of NMN and related pyridine compounds, in different cells and tissues, with picomolar sensitivity. Finally, the simultaneous quantitation of nicotinamide riboside, nicotinamide mononucleotide and NAD by means of a fluorometric, enzyme-coupled assay has been recently reported [19,20].

Nevertheless, no biosensing systems are reported for NMN. On the other hand, such in-vivo and in-vitro biosensors have been recently developed for NAD, providing a powerful tool for studying its role in cellular and extracellular microenvironment [21,22,23,24]. Development of a biosensor able to quantify NMN in subcellular compartments and biological fluids would help to clarify several aspects of NMN homeostasis and contribute to shedding light on the mechanisms of its pharmacological action.

As a first step, the designing of a biosensor needs the identification of a suitable molecular recognition element (MRE) [25]. Inactivated enzymes are among the biomolecules of choice as MREs for small-molecule detection [26,27]. Indeed, it is of fundamental importance that the MRE is able to bind the substrate specifically, with a relatively high affinity (with regards to the physiological concentrations to be measured), but not to transform it into a product.

Recently, we have identified and functionally characterized the enzyme NMN deamidase from *E. coli*, also known as PncC (EC 3.5.1.42) [28]. PncC is a member of the amidohydrolase group, that catalyzes the hydrolysis of the carbamide bond in the nicotinamide moiety of NMN, yielding nicotinic acid mononucleotide (NaMN). Based on molecular docking experiments performed on the available structure of PncC from *A. tumefaciens* (atPncC, PDB code 2A9S) in our previous work, we have identified and selected for site directed-mutagenesis several highly conserved amino acids potentially involved in the catalytic activity. Kinetic analysis of the mutated proteins showed that mutants S29A, Y56A, K61Q, and R142A completely lost catalytic activity [29].

In the present work, we have focused on these PncC mutants, with the aim to investigate their suitability as molecular recognition elements, for the designing of an MNM biosensor. We found that two PncC mutants, S29A PncC and K61Q PncC, still preserve their ability to bind NMN, but at the same time they are not catalytically active. Using thermal-shift assays and steady state-fluorescence spectroscopy measurements, we investigated their affinity binding properties to NMN. In addition, we studied the effect of NaMN, and of several other NNM-related pyridines nucleotides such as nicotinic acid (Na), Nam, NR, NAD (P), and nicotinic acid adenine dinucleotide (NaAD) on the fluorescence features of the two mutated enzymes, in order to verify the specificity of their binding properties.

## 2. Results

### 2.1. S29A PncC and K61Q PncC Mutants Preserve the Ability to Bind NMN and the Dimeric Structure of the Wild-Type PncC Protein

The PncC enzyme consists of 165 residues with a molecular mass of 17.6 kDa. Our previous work identified critical amino acid residues involved in the PncC catalyzed reaction [29]. Among them, S29, Y56, K61, and R142 were found to be essential for the enzyme activity as their substitution yielded proteins that were fully inactive.

To assess if the four mutated PncC proteins were still able to bind their natural substrate, we first performed thermal shift assays in the absence and presence of 1 mM NMN. Figure 1 shows representative thermal unfolding curves of the wild type (WT) PncC and of the four PncC mutants (S29A, Y56A, K61Q, and R142A). As shown in the table (Figure 1 inset), S29A PncC and Y56A PnC mutants retained a similar thermal stability compared to the wild type enzyme, exhibiting Tm values of 85.5 and 84 °C, respectively (Tm of the wild-type is 86.1 °C). On the contrary, K61Q PncC and R142A PncC were significantly less stable, displaying Tm values of 67.5 and 52.5 °C, respectively. The Tm values of R142A PncC and Y56A PncC were not affected by the addition of 1.0 mM NMN, whereas the Tm values of wild-type PncC, S29A PncC, and K61Q PncC increased significantly in the presence of 1.0 mM NMN (by 6.1, 10.0, and 18.7 °C respectively). These results suggest that, while R142A PncC and Y56A PncC fully lost their ability to bind NMN, S29A PncC and K61Q PncC were still able to bind it.

The oligomeric states of S29A PncC and K61Q PncC mutants were determined through gel filtration chromatography experiments. As shown in Figure 2a, both PncC mutants behaved as the wild type protein, yielding a peak corresponding to a molecular mass of about 30,000 Dalton, indicating that mutations did not affect the dimeric structure of the proteins.

The integrity of the two PncC mutated protein structures were investigated also by steady-state fluorescence measurements, performed at 25 °C. In particular, we evaluated the intrinsic fluorescence emission due to the protein tryptophan residues (W34, W48, W121, and W158). The obtained data show that the fluorescence emission spectrum of WT PncC presents a maximum peak centered at 332 nm (Figure 2b). This position is blue-shifted with respect to the emission maximum of N-acetyltryptophanylamide (NATA) centered at 350 nm (data not shown), suggesting that the tryptophanyl residues of PncC at 25 °C are partially buried and/or are located in un-relaxed microenvironments. No shifts of the fluorescence emission maximum peaks were detected for the S29A PncC and K61Q PncC mutants, suggesting that these mutations do not affect the Trp residues exposition to solvent (Figure 2b).

### 2.2. S29A PncC and K61Q PncC Bind NMN in the Micromolar Range of Concentrations

In order to determine the binding affinity constants (*K_d_*) of S29A PncC and K61Q PncC for NMN, we performed thermal shift assays and steady-state fluorescence experiments, at increasing concentrations of NMN.

First, we analyzed the thermal stability of the two proteins at nucleotides concentrations ranging from 10 to 900 µM.

By plotting the percentage of unfolded proteins as a function of NMN concentrations, *K_d_* values of 64 and 91 µM were calculated for S29A PncC and K61Q PncC, at 86 and 72 °C, respectively (Figure 3a,b). A two-step denaturation profile was observed for the mutant S29A in the presence of NMN (Figure 3a), which might be due to the ligand-induced dissociation of the dimer, followed by denaturation of the monomers.

The effect of the binding of NMN to S29A PncC and K61Q PncC mutants was also investigated by fluorescence spectroscopy. Titration experiments were carried out in the range 0.18–96 µM of NMN. As shown in Figure 3c,d, in the presence of NMN, no shift in the emission maximum wavelength was observed and the fluorescence emission intensity was quenched at increasing concentrations of NMN in both mutants. By plotting the variation of the fluorescence emission intensity at 332 nm as a function of NMN concentration, apparent *K_d_* values of 37 µM (R^2^ = 0.9998) and 125 µM (R^2^ = 0.9968) were determined at 25 °C for S29A PncC and K61Q PncC, respectively. Hill coefficient values close to 1 were obtained for both proteins (inset table Figure 3), indicating an independent binding of NMN.

These data confirm that both mutants are able to bind to NMN showing that the *K_d_* values, obtained with the two different methodologies, are in the micromolar range. In addition, these results suggest that NMN induces conformational changes of the protein structures, influencing the Trp fluorescence emission.

### 2.3. Evaluation of the PncC Mutants Capability to Bind NaMN

To rule out possible interference of the presence of NaMN, the product of the PncC catalyzed reaction, thermal shift, and steady-state fluorescence experiments similar to those described for NMN, were conducted in the presence of NaMN.

Both thermal shift and steady state fluorescence measurements (illustrated in Figure 4a,c) showed that S29A PncC is able to bind NaMN. Indeed, the apparent *K_d_* resulted to be 59 µM, at 89 °C, from thermal shift assays, and 11.7 µM, at 25 °C from intrinsic Trp fluorescence measurements. These results were similar to results obtained by steady state fluorescence on WT PncC (not shown), indicating that the S29A PncC mutation does not interfere with the enzyme ability to bind neither the substrate nor the enzymatic product. As observed with NMN (Figure 3a), also with NaMN the mutant proteins display a two-step denaturation profile (Figure 4a), suggesting that both ligands might induce dissociation of the dimer before the monomer’s denaturation.

On the contrary, the results obtained on K61Q PncC mutant with both techniques showed that it does not significantly interact with NaMN (Figure 4b,d). Indeed, by plotting the percentage of unfolded protein (Figure 4b) or the variations of the fluorescence emission peak (not shown) as a function of NaMN concentration, it was not possible to determine *K_d_*.

From these results, we concluded that, while S29A PncC binds both to NaMN and NMN, K61Q PncC has a marked preference of binding for NMN, thus, being a possible candidate for an NMN sensor.

### 2.4. Effects of Other NMN-Related Metabolites on Intrinsic Trp Fluorescence

In our previous work [28], we demonstrated that NMN is the specific substrate for PncC since several other pyridine-nucleotides were not modified by the enzyme. Nevertheless, the issue of whether the enzyme is able to bind other molecules, without modifying them, and the effect of the introduced mutations on the capacity to bind metabolites in a non-specific way, has not been investigated. Therefore, we have addressed the ability of S29A PncC and K61Q PncC to bind the following molecules, structurally related to NMN: Na, Nam, NR, NAD, NADP, and NaAD (structures and molecular weight are reported in Figure 5).

In Figure 6, the normalized fluorescence intensity emission values of S29A PncC and K61Q PncC are shown as a function of increasing concentrations of the different molecules. Values obtained with equivalent volumes of buffer, and with NMN and NaNM are also reported for comparative purposes. The corresponding percentages of signal reduction for each metabolite concentration, compared to buffer, are reported in Appendix A.

As for S29A PncC, the results showed that, with the exception of NMN and NaMN, curves of all other metabolites were nearly overlapping with the curve obtained with buffer alone, at least at low concentrations. Nevertheless, increasing the molecule concentrations, a quenching of the fluorescence is evident, exceeding the threshold of 4% (see Appendix A), for all the molecules except Na, that does not cause a fluorescence decrease (Figure 6a and Appendix A). The effect on fluorescence emission is higher for larger molecules containing the adenine group, such as NAD, NADP, and NaAD, inducing a maximum fluorescence quenching of 7.9, 9.6, and 6.7%, respectively, in comparison to NaMN and NMN values of 52.8 and 55.2%, respectively.

Interestingly, results obtained on K61Q PncC highlighted that several molecules, with the exception of NMN, have a substantially lower effect on the tryptophanyl fluorescence emission intensity, in comparison to the effect on S29A PncC (see Appendix A). Indeed, the presence of Nam, NAD, or NR causes a very low fluorescence quenching (<2%), even at 96 µM concentration, whilst Na and NADP contribute to decrease the emission of 4.5 and 5.1%, compared to the corresponding buffer. At the same time, although a quenching of fluorescence emission is observed in the presence of NaAD at high concentrations, it is limited to about 8% level. Finally, it is worth to notice that the interaction of NaMN with K61Q PncC resulted in a 10.68% decrease in fluorescence emission, compared to a value of 53.3%, found in presence of NMN.

To confirm the obtained data and to exclude non-specific binding of NMN and NaMN to proteins, as control we performed fluorescence steady-state experiments in the same conditions with a protein such as the glutamine binding protein (GlnBP), a small protein involved in the glutamine intake in *E. coli*. The obtained data show a quenching of fluorescence of 4.3% in the presence of 96 µM NMN or NaMN (Appendix A). Also, a weak reduction of fluorescence was observed at high concentration of NAD (6.6%), NADP (4.8%), and NaAD (6.3%), while Nam, NR, and NA do not affect the steady-state fluorescence of GlnBP (data not shown).

## 3. Discussion

In the last years, there is an increasing interest in the detection and quantification of NMN in subcellular compartments and biological fluids. Indeed, the development of an NMN-biosensor would help to clarify its roles in cellular metabolism and disease. The identification of an appropriate MRE is the first crucial step for the designing of effective biosensors [25]. Once biochemically characterized, the sensing element can be fused to a probe providing a fluorescence readout, including fluorescent organic molecules, nanoparticles, proteins, or a combination of these [26,27]. Consequently, the binding of the ligand to the specific MRE results in a conformational variation that can be detected by fluorescence measurements. Examples of these biosensors have been recently reported for the NAD detection. They are based on NAD-binding domains of bacterial enzymes as MREs, fused to circularly permuted fluorescent proteins [22,23,24].

To the scope of characterizing a suitable NMN-recognition element, in this work, we investigated the binding affinity and the specificity toward NMN of two mutated variants of the *E. coli* PncC enzyme. In particular, among the four mutated PncC previously identified [29], we selected the S29A PncC and K61Q PncC, since they do not exhibit catalytic activity [29], and, at the same time, they retain the ability to bind NMN.

Indeed, we observed that in the presence of 1 mM of NMN there is an increase of the melting temperature of S29A PncC, K61Q PncC, and WT PncC.

On the contrary, Y56A PncC and R142A PncC, showed unvaried unfolding curves both in the absence and in the presence of NMN. These results agree with our previous structural studies [29], where it was demonstrated that amino acids Y56 and R142 are involved in the maintenance of the active site architecture of the enzyme. Indeed, far-UV CD analyses performed on Y56A PncC and R142A PncC revealed significant conformational changes with respect to the wild type protein. On the other hand, S29 and K61 were identified as the two Ser/Lys residues utilized by PncC to catalyze its reaction. Recently, this has been also described for the PncC from *A.*
*tumefaciens* [30].

Further indications that S29A and K61Q substitutions do not affect the protein conformational structure derive from experiments confirming that mutants retain the dimeric structure of the wild type protein, as well as the same absorbance and steady-state fluorescence emission spectra.

Apparent dissociation constants values of NMN were determined for both the mutated proteins by thermal shift assays and by steady-state fluorescence measurements. The results showed that for S29A PncC the *K_d_* values range from 64 µM, as found by thermal shift, to 37 µM, as found by intrinsic fluorescence analysis. Analogously, *K_d_* of NMN interaction with K61Q PncC ranges from 91 to 125 µM. The slight discrepancy in the *K_d_* values obtained from the two approaches could be explained by inherent differences between the two utilized techniques [31,32], and/or by the different ranges of the ligand amounts used for the titration experiments.

Steady-state fluorescence emission of the intrinsic Trp residues, is influenced by conformational changes of the protein structure, that can be used to track folding/unfolding, oligomerization, and also protein/ligand interaction [33]. The PncC protein sequence includes four tryptophanyl residues (W34, W48, W121, and W158). From in silico structural analysis, none of these four amino acids resulted to be directly involved in the protein active site [29]. Therefore, the NMN dose-dependent reduction of the tryptophanyl fluorescence, in both PncC mutated proteins, is likely explained by ligand-induced conformational changes of the protein structure, influencing the Trp micro-environment [33].

The micromolar *K_d_* values obtained for both mutants are higher than NMN concentrations in biological fluids and tissues [8,9], suggesting that the mutants would respond in a proportional manner to changes in cellular NMN levels. As an example, for PncC K61Q mutant, based on the *K_d_* value of 125 µM, we calculated that for NMN concentrations ranging from 1 to 50 µM, the occupancy of the mutant would range from 0.8% to 29%. We also tested the ability of the two mutated proteins to bind NaMN, the product of the enzymatic reaction catalyzed by PncC. Intracellular NaMN levels are generally from 20- to 40-times lower than NMN levels, depending on the tissue type [16]. Importantly, we found a substantial difference in the ability of the two mutants to bind NaMN. Indeed, while S29A PncC binds NaMN with an affinity comparable to that for NMN, K61Q PncC does not significantly interact with it.

Finally, to rule out the possibility that the mutated proteins bind other NMN-related nucleotides, we performed steady-state fluorescence measurements in the presence of several other pyridine nucleotides related to NMN, including Na, NR, Nam, NAD, NADP, and NaAD. The results showed that a reduction of the S29A PncC Trp fluorescence emission was observed, at the highest concentration assayed (96 µM), also in presence of NAD, NADP, or NaAD. On the other side, these NMN-related nucleotides, had a lower effect on the Trp fluorescence emission of K61Q PncC, compared to the results obtained with S29A PncC. In particular, the results suggested that a residual binding ability was present to NADP and NaAD, but, overall, they induced a fluorescence emission reduction lower than those induced on the structure of S29A PncC. The small decrease in PncC protein fluorescence emission observed at the highest concentrations of NaAD, NAD, and NADP could be due to the partial absorption of the excitation light by the adenine group of these compounds. Actually, these compounds show a weak absorbance at 295 nm, that may interfere with the Trp excitation process. This phenomenon can be excluded for NMN and NaMN, showing a very limited (at least 10-fold lower) adsorption at 295 nm, compared to the adenine containing nucleotides.

We conclude that, while S29A PncC is not able to recognize specifically NMN, especially because it cannot discriminate between NMN and NaMN, K61Q PncC could be a promising probe for the designing of an NMN biosensor. In fact, K61Q PncC is an inactivated enzyme, that is still able to bind NMN in the µM range, and does not significantly interact with NaMN and other related metabolites.

Nevertheless, further studies are needed for a biosensor design, including a deeper investigation on possible ligand-induced conformational changes in the protein structure, as well as the analysis of the protein stability in different conditions that may influence the biosensor performance, such as pH, temperature, ionic strength and solvent polarity.

## 4. Materials and Methods

### 4.1. Chemicals

NMN, NaNM, Na, Nam, NR, NAD, NADP, NaAD, and TCEP were purchased from Sigma (Sigma-Aldrich Milano, Italy). Other chemicals were grade A substances obtained from VWR (VWR International SRL Milano, Italy). Stock solutions of the nucleotides were prepared at 100 µM concentration in Milli-Q water and stored at –20 °C.

### 4.2. Expression, Purification and Gel Filtration Chromatography of Recombinant PncC Wild Type, S29A, and K61Q

Recombinant proteins were expressed and purified as previously reported [28]. Briefly, pCA24N-pncC, and its mutagenized plasmids were expressed in *E. coli* BL21(DE3) cells (Novagen, Podenzano, Italy). Proteins were purified from the soluble fraction of bacterial lysates, by HisTrap HP column (GE Healthcare, Chicago, IL, USA) equilibrated with 50 mM HEPES/NaOH pH 7.5, 0.3 M NaCl buffer, and eluted with an imidazole linear gradient. The identification of the fractions containing the protein of interest and its purity was carried out by SDS-PAGE. The pools collected were dialyzed against 50 mM HEPES/NaOH buffer, pH 7.5, 0.3 M NaCl, and then characterized.

Native MW of proteins was determined by gel filtration on a Superose 12 HR 10/30 column (GE Healthcare, Chicago, IL, USA) eluted with 50 mM HEPES pH 7.5, 0.3 M NaCl, 1 mM TCEP.

### 4.3. Fluorescence-Based Thermal Shift Binding Assay

Protein melting temperature (Tm) was determined by fluorescence-based thermal shift assay using fluoroprobe SYPRO Orange dye (Invitrogen, Waltham, MA, USA). The assay was performed in a RotorGene 3000 real-time PCR (Corbett, Biocompare, San Francisco, CA, USA). The assay mixture contained 50 mM Hepes/NaOH, pH 7.5, 300 mM NaCl, 1 mM TCEP, 1 μM protein, Sypro orange at 10-fold dilution from a 5000-fold stock, and the various ligands at the tested concentrations, in a final volume of 25 μL. The mixtures were pre-heated at 40 °C for 5 s, then heated to 50 °C in 1 s, and finally heated from 50 to 99 °C with a heating-up rate of 0.5 °C every 5 s. The fluorescence intensity was measured with Ex/Em: 470/585 nm. The melting temperature was determined by taking the maximum point of the slope-derivative peak. *K_d_* values were determined by plotting the percentage of unfolded proteins as a function of nucleotide concentration at a fixed temperature, after normalization of the unfolding curves [34]. Fitting of the data was performed using the equations reported in [34]. Three independent experiments were carried out and values were reported as the means ± standard deviation.

### 4.4. Steady-State Fluorescence Spectroscopy

Steady-state measurements were performed using an ISS_K2 fluorometer (ISS, Urbana-Champaign, IL, USA), equipped with a one-cell temperature-controlled sample holder. Steady-state fluorescence emission spectra of WT PncC and of the two mutants, were acquired setting excitation value at 295 nm and recording between 310 and 410 nm at 1.0 nm intervals. Excitation and emission slit widths were fixed at 0.5 and 1.0 nm, respectively. In binding experiments, the fluorescence spectra were carried out through incubation of proteins at a final concentration of 3 µM (absorbance about 0.04 OD) in 50 mM HEPES pH 7.4, 300 mM NaCl, and 1 mM TCEP, with increasing concentrations of metabolites (NMN, NaMN, Na, Nam; NR, NAD, NADP, and NaAD) or with an equal volume of buffer, under continuous stirring, at 25 °C. The nucleotides were added to reach the indicated final concentration (from 0.18 to 96 µM) with an incubation interval of 10 min. Triplicate readings were done for each experimental point. Experiments were repeated three times (for NMN and NaMN) or twice (for all the other metabolites). Controls experiments were performed using the *E. coli* Glutamine Binding Protein (GlnBP, 3µM), with increasing concentration of NMN or NaMN, at the same conditions.

Experimental data were processed using the OriginPro 8.0 (OriginLab Corporation, Northampton, MA, USA). A non-linear curve analysis (Hill1 plot) was used for plotting the F0/F values in function of the metabolite concentration, to obtain the *K_d_* value.

Percentages of reduction of fluorescence intensities compared to buffer, were calculated as follow: (normalized fluorescence in presence of analyte at a given [µM]/normalized fluorescence in presence of corresponding volume of buffer) × 100.

### 4.5. PncC Protein Occupancy Calculation

Percentage of the K61Q protein that would exist in complex with NMN in a cellular setting was calculated based on the *K_d_* value, by using the equation:$$\%=\frac{100}{1+\frac{K_{d}}{\begin{array}{c} \left[NMN\right] \end{array}}}$$

## Figures and Tables

**Figure 1 ijms-22-06334-f001:**
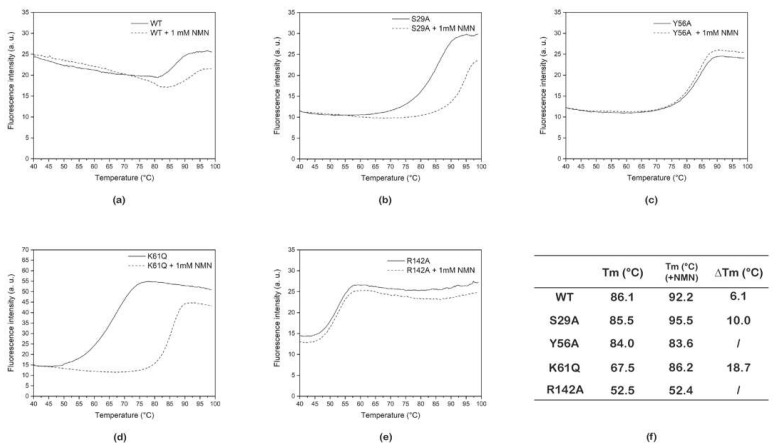
Thermal stability assays of WT PncC (**a**) and its mutants (**b**–**e**), in the absence and in the presence of NMN. Representative denaturation curves obtained for wild-type and mutated PncC proteins in the absence (continuous line) and in the presence (dashed line) of 1 mM NMN. In the table (**f**), the melting temperature values of the proteins are reported.

**Figure 2 ijms-22-06334-f002:**
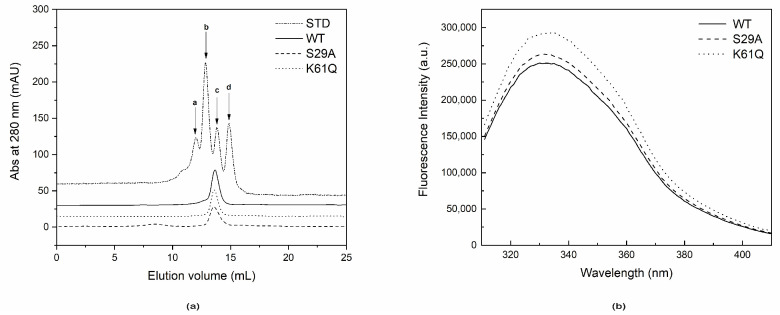
Evaluation of the oligomeric state and of intrinsic Trp fluorescence emission of WT PncC, S29A PncC, and K61Q PncC. (**a**) Elution profiles from gel filtration chromatography of wild-type PncC (45 µg), K61Q PncC, and S29A PncC mutants (30 µg) and protein standards (a, bovine serum albumin, 66 kDa; b, ovalbumin, 45 kDa; c, carbonic anhydrase, 20 kDa; d, cytochrome C, 12.3 kDa). (**b**) Representative steady-state fluorescence emission spectra of the proteins (3 µM) (Ex wavelength = 295 nm).

**Figure 3 ijms-22-06334-f003:**
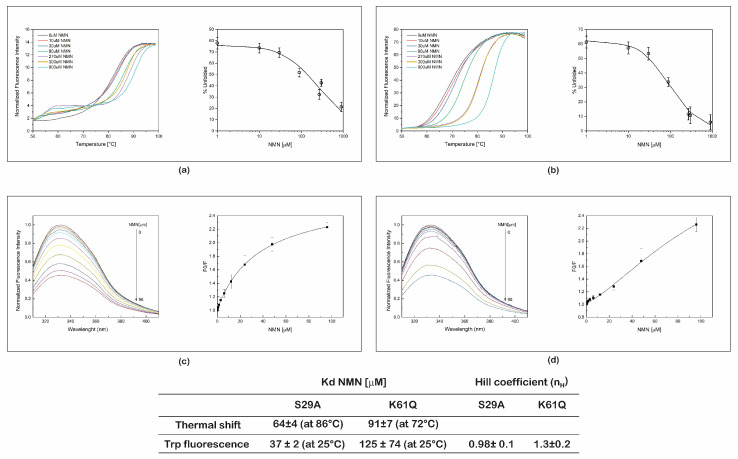
Determination of the apparent dissociation constants of NMN to S29A PncC and K61Q PncC. (**a**,**b**) Representative normalized unfolding curves of S29A PncC and K61Q PncC in the presence of various amounts of NMN, with the corresponding plots of the percentage of unfolded protein as a function of NMN concentrations at 86 and 72 °C, respectively. (**c**,**d**) Representative fluorescence emission spectra of S29A PncC and K61Q PncC in the presence of NMN at increasing concentrations, with the corresponding plots of F0/F as a function of NMN concentration. In the plot images, mean values (of % of unfolded protein or F0/F) ± SD of triplicate experiments are reported. In the inset table, the *K_d_* values obtained from both techniques are reported.

**Figure 4 ijms-22-06334-f004:**
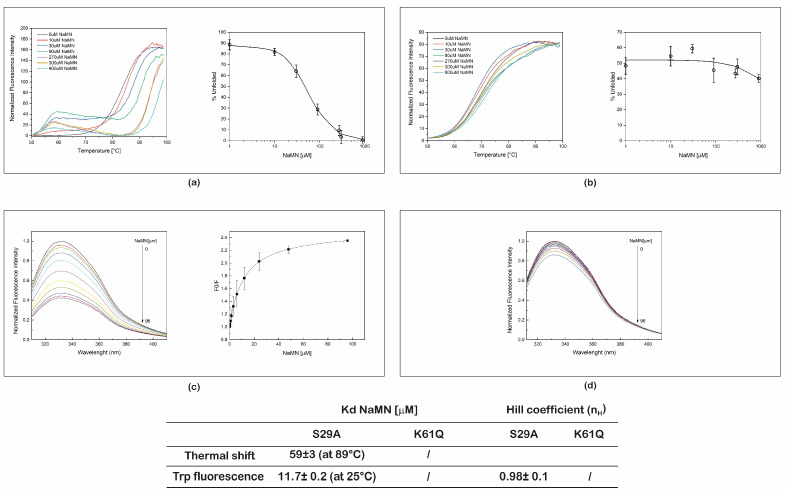
Determination of the apparent dissociation constants of NaMN to S29A PncC and K61Q PncC. (**a**,**b**) Representative normalized unfolding curves of S29A PncC and K61Q PncC in the presence of various amounts of NaMN, with the corresponding plots of the percentage of unfolded protein as a function of the nucleotide concentration, at 89 and 70 °C, respectively. (**c**,**d**) Representative fluorescence emission spectra of S29A PncC and K61Q PncC in the presence of NaMN at increasing concentrations, with the corresponding plots of F0/F as a function of NaMN concentration. In the plot images, mean values (of % of unfolded protein or F0/F) ± SD of triplicate experiments are reported. In the inset table, the *K_d_* values obtained from both techniques are reported.

**Figure 5 ijms-22-06334-f005:**
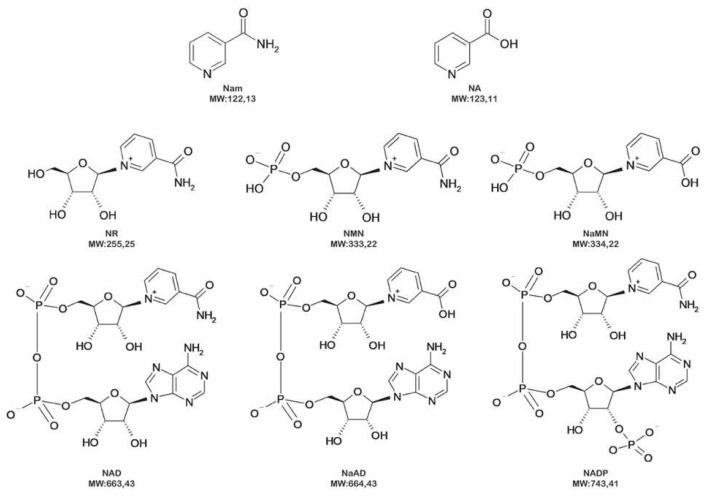
Chemical structures of the NMN-related nucleotides included in this study.

**Figure 6 ijms-22-06334-f006:**
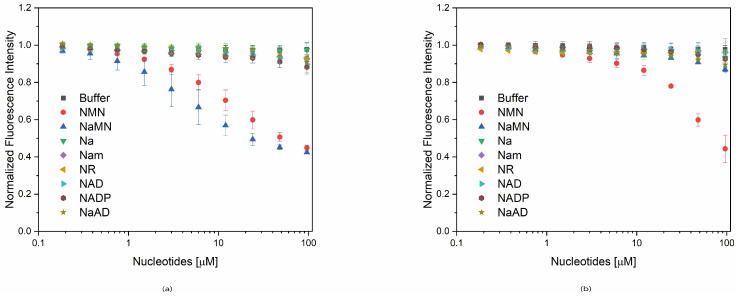
Effects of NMN-related nucleotides on the steady-state fluorescence of S29A and K61Q PncC proteins. Variation of the maximum fluorescence emission intensity values of S29A PncC (**a**) and K61Q PncC (**b**) as function of the increasing concentration of the different molecules. Normalized fluorescence mean values ± SD of triplicate experiments (buffer, NMN, and NaMN) or of duplicate experiments (Na, Nam, NR, NAD, NADP, and NaAD) are reported.

## Data Availability

The datasets corresponding to the current study are available from the corresponding author upon request.

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
