# Peer review of "Characterization of Two NMN Deamidase Mutants as Possible Probes for an NMN Biosensor"

_ijms, 2021, doi:10.3390/ijms22126334_

Round 1

Reviewer 1 Report

Camarca et al characterized 4 inactive mutants of NMN deaminase with the intention of designing an NMN biosensor. The mutations were made in a previous study of the same group. Here, the authors assessed the affinity and specificity of 4 mutants for MNM and analogous metabolites.

The results are original and interesting but some revisions are necessary to facilitate the comprehension of the manuscript and to correct some errors and mis-/over-interpretations.

Because the affinity and specificity of the biosensor is as important as the non-transformation of the molecule to be detected, I suggest modifying line 71 as follows: ...MRE is able to bind the substrate specifically and with a relatively high affinity (with regards to the physiological concentrations to be measured), but not to transform it.

Line 80: Y56A instead of Y65A.

Sentence starting at Line 105 to be replaced with: The Tm of R142A PncC and Y56A PncC was not affected by the addition of 1 mM NMN, whereas the Tm of wild-type PncC, S29A PncC and K61Q PncC increased significantly (by 6.1, 10.0 and 18.7°C respectively).

Figure 1: The thermal stability of the WT enzyme is not represented over the same temperature range. Is this due to a 2-step denaturation of the protein, as seen, but not discussed, in Figure 4a? A two-step denaturation profile can be explained by the dissociation of the dimer followed by the denaturation of the monomers. In all cases, Figure 1a should be shown from 40 to 100°C.

Figure 1 (and the rest of the manuscript): It would be easier for the reader if the panels and the results in the table are presented in the order of the amino acid sequence, i.e. in this order: WT, S29A, Y56A, K61Q and R142A.

Inset Table of Figure 1: Please indicate the unit of Tm and Delta Tm. Use the same number of decimal places for all digits.

Figure 2: Keep the same order as recommended previously (WT, S29A and K61Q).

Figure 2: There are no molecular weight standards shown on the elugram and therefore it is difficult to be convinced by the advertised value of the molecular weight of the protein. In the Methods section, it is mentioned that standards are 66, 45 and 30 kDa). Since the PncC monomer has an expected molecular mass of 17.6 kDa, a standard with a lower molecular weight (around 15 kDa) would be useful to distinguish the PncC monomer from the PncC dimer.

Paragraph starting Line 138 and further: It is not correct to use the dependence of deltaTm on ligand concentration to determine the binding affinity because the binding constant is temperature dependent. A more correct way to quantify the binding affinity from thermal denaturation curves is to use the approach proposed by Bai et al. (doi: 10.1038/s41598-018-37072-x.). This leads to an estimate of the binding affinity at a given temperature. However, it may be difficult to use this technique for the K61Q mutant because the melting temperature varies to a large extent. At the very least, the authors should state that the apparent Kd that they measure by this technique is not precise (for the reason evoked previously). This may explain the difference between the Kd values measured from TSA experiments and fluorescence titration assays.

Table of Figure 3: Please remove the decimals from the Kd values and SD because the precision is not high enough: 189 +/- 6.7 and 125 +/- 74. Give and discuss the value of the Hill coefficients measured.

Figure 4a: With the WT protein, the traces of fluorescence as a function of temperature in the presence of NaMN are not conventional. Denaturation appears to be biphasic. Please discuss this aspect. Is this due to the dissociation of the dimer induced by NaMN? This could be verified by the SEC-MALS technique (or simply SEC if the separation resolution is sufficient) or by SAXS for example.

Figure 4 Table: 11.7 +/- 0.2 (instead of 11.71 +/- 0.2).

The small decrease in PncC fluorescence observed at the highest concentrations of NaAD, NAD and NADP may be due to the partial absorption of the excitation light by the adenine group of these compounds. To a much lesser extent, nicotinamide can also absorb at 295 nm at high concentrations. A measurement of the absorption spectra of these metabolites at 96 µM in the buffer should be sufficient to conclude whether they absorb part of the emission light (at 295 nm) or not. Please comment on this aspect. Monitoring with GlnBP can also be performed with metabolites containing adenine.

Tables 1 and 2 can be presented in the supplementary information.

Rather, Figure S1 should appear in the main manuscript. There are many corrections/errors in this Figure. (1) Please have a homogeneous representation of the molecules (always in the same orientation). (2) Represent molecules starting from the simplest (Na or Nam) to the most complex (NADP). (3) NR is missing. (4) The molecular weight of NaAD is 663.40, not 745.39. (5) Moreover, it is the NaAD phosphorylated in 2' (NaADP) which is represented in the figure instead of NaAD. (6) The last molecule is NADP (and not NaAD). The molecular weight of NADP is 743.41.

Discussion: It seems that, given the physiological concentrations of NMN, the affinity of the K61Q PncC for NMN is not sufficient for a biosensor. How to improve this? What is the strategy for developing an NMN biosensor from an inactive PncC mutant (output signal)?

The quality of the figures must be greatly improved.

Author Response

Response to Reviewer 1 Comments

Camarca et al characterized 4 inactive mutants of NMN deaminase with the intention of designing an NMN biosensor. The mutations were made in a previous study of the same group. Here, the authors assessed the affinity and specificity of 4 mutants for MNM and analogous metabolites.

The results are original and interesting but some revisions are necessary to facilitate the comprehension of the manuscript and to correct some errors and mis-/over-interpretations.

The manuscript has been modified. In particular:

Point 1: Because the affinity and specificity of the biosensor is as important as the non-transformation of the molecule to be detected, I suggest modifying line 71 as follows: ...MRE is able to bind the substrate specifically and with a relatively high affinity (with regards to the physiological concentrations to be measured), but not to transform it.

Response 1: Done

Point 2: Line 80: Y56A instead of Y65A.

Response 2: Done

Point 3: Sentence starting at Line 105 to be replaced with: The Tm of R142A PncC and Y56A PncC was not affected by the addition of 1 mM NMN, whereas the Tm of wild-type PncC, S29A PncC and K61Q PncC increased significantly (by 6.1, 10.0 and 18.7°C respectively).

Response 3: Done

Point 4: Figure 1: The thermal stability of the WT enzyme is not represented over the same temperature range. Is this due to a 2-step denaturation of the protein, as seen, but not discussed, in Figure 4a? A two-step denaturation profile can be explained by the dissociation of the dimer followed by the denaturation of the monomers. In all cases, Figure 1a should be shown from 40 to 100°C.

Response 4: In the new Fig. 1a the denaturation profile from 40°C to 100°C of the wild type enzyme is displayed. It shows that the wild type protein does not exhibit a two-step denaturation curve. Figure 4a does not refer to the wild type protein, but to S29A mutant protein. In the revised version of the manuscript we have discussed the two-step denaturation profile of this mutant protein in the presence of NMN (Figure 3a) or NaMN (Figure 4a).

Point 5: Figure 1 (and the rest of the manuscript): It would be easier for the reader if the panels and the results in the table are presented in the order of the amino acid sequence, i.e. in this order: WT, S29A, Y56A, K61Q and R142A.

Response 5: Done

Point 6: Inset Table of Figure 1: Please indicate the unit of Tm and Delta Tm. Use the same number of decimal places for all digits.

Response 6: Done

Point 7: Figure 2: Keep the same order as recommended previously (WT, S29A and K61Q).

Response 7: Done

Point 8: Figure 2: There are no molecular weight standards shown on the elugram and therefore it is difficult to be convinced by the advertised value of the molecular weight of the protein. In the Methods section, it is mentioned that standards are 66, 45 and 30 kDa). Since the PncC monomer has an expected molecular mass of 17.6 kDa, a standard with a lower molecular weight (around 15 kDa) would be useful to distinguish the PncC monomer from the PncC dimer.

Response 8: In the revised version of the manuscript we have included in the figure the elution profile of the protein standards, including cytochrome C (MW 12.3 kDa). This confirms the dimeric structure of PncC.

Point 9: Paragraph starting Line 138 and further: It is not correct to use the dependence of deltaTm on ligand concentration to determine the binding affinity because the binding constant is temperature dependent. A more correct way to quantify the binding affinity from thermal denaturation curves is to use the approach proposed by Bai et al. (doi: 10.1038/s41598-018-37072-x.). This leads to an estimate of the binding affinity at a given temperature. However, it may be difficult to use this technique for the K61Q mutant because the melting temperature varies to a large extent. At the very least, the authors should state that the apparent Kd that they measure by this technique is not precise (for the reason evoked previously). This may explain the difference between the Kd values measured from TSA experiments and fluorescence titration assays.

Response 9: We thank the referee for this suggestion. We have calculated the Kd values at selected, fixed temperatures according to Bai et al.

In Fig. 3a, Fig. 3b, Fig. 4a, Fig. 4b the normalized fluorescence curves and the corresponding plots of the percentage of unfolded proteins as a function of ligand concentrations are displayed. The analysis revealed that NaMN did not significantly affect the denaturation curve of K61Q mutant protein (Fig. 4a). This is in agreement with the results obtained with the Trp fluorescence assay (Fig. 4d). The manuscript text has been modified accordingly (lines 156-158, 191-193 and 213-217).

Point 10: Table of Figure 3: Please remove the decimals from the Kd values and SD because the precision is not high enough: 189 +/- 6.7 and 125 +/- 74. Give and discuss the value of the Hill coefficients measured.

Response 10: Decimals have been removed. The Hill values have been reported and discussed (Inset tables of figures 3 and 4).

Point 11: Figure 4a: With the WT protein, the traces of fluorescence as a function of temperature in the presence of NaMN are not conventional. Denaturation appears to be biphasic. Please discuss this aspect. Is this due to the dissociation of the dimer induced by NaMN? This could be verified by the SEC-MALS technique (or simply SEC if the separation resolution is sufficient) or by SAXS for example.

Response 11: Fig. 4a refers to the S29A mutant. Indeed, denaturation appears to be biphasic for this protein in the presence of NMN (Fig. 3a), as well as NaMN (Fig. 4a). In the revised version of the manuscript we have discussed this aspect (lines 158-160 and 197-199).

Point 12: Figure 4 Table: 11.7 +/- 0.2 (instead of 11.71 +/- 0.2).

Response 12: Done

Point 13: The small decrease in PncC fluorescence observed at the highest concentrations of NaAD, NAD and NADP may be due to the partial absorption of the excitation light by the adenine group of these compounds. To a much lesser extent, nicotinamide can also absorb at 295 nm at high concentrations. A measurement of the absorption spectra of these metabolites at 96 µM in the buffer should be sufficient to conclude whether they absorb part of the emission light (at 295 nm) or not. Please comment on this aspect. Monitoring with GlnBP can also be performed with metabolites containing adenine.

Response 13: We thank the referee for this suggestion. We have measured the absorption spectra of all nucleotides at a concentration of 96 µM. The obtained OD values at 295nm are reported below. As pointed by the referee, we cannot exclude that the quenching of fluorescence in the presence of NAD, NADP and NaAD can be due to internal absorption of the excitation light.

In addition, we have measured the Trp-emission curves of GlnBP in the presence of metabolites, as indicated, showing that NAD, NADP and NaAD also determine a weak fluorescence quenching of GlnBP.

We have added the comments on this aspect in the results section as well as in the discussion section (lines 256, 274-276, 345-351).

Nucleotide in buffer

Adsorbance (OD295nm)

Na (96uM)

0.000

Nam (96uM)

0.000

NR (96uM)

0.000

NMN (96uM)

0.002

NaMN (96uM)

0.001

NAD (96uM)

0.024

NADP (96uM)

0.019

NaAD (96uM)

0.027

Point 14: Tables 1 and 2 can be presented in the supplementary information.

Response 14: We have moved table 1 and table 2 in the supplementary materials.

Point 15: Rather, Figure S1 should appear in the main manuscript. There are many corrections/errors in this Figure. (1) Please have a homogeneous representation of the molecules (always in the same orientation). (2) Represent molecules starting from the simplest (Na or Nam) to the most complex (NADP). (3) NR is missing. (4) The molecular weight of NaAD is 663.40, not 745.39. (5) Moreover, it is the NaAD phosphorylated in 2' (NaADP) which is represented in the figure instead of NaAD. (6) The last molecule is NADP (and not NaAD). The molecular weight of NADP is 743.41.

Response 15: We have modified the figure and moved it in the main manuscript.

Point 16: Discussion: It seems that, given the physiological concentrations of NMN, the affinity of the K61Q PncC for NMN is not sufficient for a biosensor. How to improve this? What is the strategy for developing an NMN biosensor from an inactive PncC mutant (output signal)?

Response 16: In the revised version of the manuscript, we report the percentage of protein that would be occupied by NMN in physiological conditions, as determined by considering the Kd value of 125 µM and NMN concentrations ranging from 1 to 50 µM. We found that the occupancy of the protein in such conditions would range from 0.8% to 29%, indicating that the protein would respond in a proportional manner to changes in cellular NMN levels. Therefore, the affinity of K61Q PncC for NMN seems to be sufficient for a biosensor at least at these cellular concentrations.

In the discussion section we have briefly described some strategies which have been successfully adopted for the development of fluorescence-based NAD biosensors and which might be useful for our scope (lines 283-289 and lines 357-360).

Point 17: The quality of the figures must be greatly improved.

Response 17: All the figures have been prepared at a resolution of 300 dpi, according to the journal recommendations.

Reviewer 2 Report

The manuscript by Camarca et al. describes a set of experiments to develop a NMN sensor based on the construction of a catalytically inactive mutant of NMN deamidase (PncC from A. tumefaciens).  There are positives about this manuscript, as listed below.

  1. The manuscript is well-written. In particular, the authors do a very good job of describing the need for NMN biosensor.
  2. The experiments are well done with proper controls included.
  3. The conclusions are supported by the data.

However, this manuscript suffers one a very serious drawback.  This reviewer would rate the work as marginally successful.   The authors identify two PncC mutants that bind NMN, S29A PncC and K61Q PncC.  The S29A PncC mutant does not discriminate well between NMN and NaNM (nicotinic acid mononucleotide).  The other mutant, K61Q PncC, does discriminate between NMN and NaNM, but the differences in binding affinity are only 8-9-fold.  In addition, the Kd value for the binding of NMN to K61Q PncC is in the range of 100-200 µM, which is not “high affinity” as stated by the authors.

It is not clear that the K61Q PncC mutant could function as a successful NMN biosensor because the Kd value is not sufficient low and the 8-9-fold discrimination between NaNM and NMN is sufficiently high.  The authors need to comment on these points in the discussion.  In fact, the authors could use the measured Kd values for NMN and NaNM and the cellular concentrations of NMN and NaNM to calculate the occupancy of K61Q PncC with each metabolite.  Basically, calculate the percentages of K61Q PnCC that would exist as K61Q PnCC-NMN complex and the K61Q PnCC-NaNM complex.  If the calculations reveal a high percentage of the K61Q PncC-NMN complex, then the authors could argue the value of their work.   With a Kd value of 100-200 µM, this reviewer suspects that percentage K61Q PnCC-NaNM complex will be low, particular if the cellular concentrations of NMN are in the 100 µM range.

Without this analysis, this work seems more of a progress report than a successful study.  

Author Response

The manuscript by Camarca et al. describes a set of experiments to develop a NMN sensor based on the construction of a catalytically inactive mutant of NMN deamidase (PncC from A. tumefaciens). There are positives about this manuscript, as listed below.

  1. The manuscript is well-written. In particular, the authors do a very good job of describing the need for NMN biosensor.
  2. The experiments are well done with proper controls included.
  3. The conclusions are supported by the data.

However, this manuscript suffers one a very serious drawback. This reviewer would rate the work as marginally successful. The authors identify two PncC mutants that bind NMN, S29A PncC and K61Q PncC. The S29A PncC mutant does not discriminate well between NMN and NaNM (nicotinic acid mononucleotide). The other mutant, K61Q PncC, does discriminate between NMN and NaNM, but the differences in binding affinity are only 8-9-fold. In addition, the Kd value for the binding of NMN to K61Q PncC is in the range of 100-200 µM, which is not “high affinity” as stated by the authors.

It is not clear that the K61Q PncC mutant could function as a successful NMN biosensor because the Kd value is not sufficient low and the 8-9-fold discrimination between NaNM and NMN is sufficiently high. The authors need to comment on these points in the discussion. In fact, the authors could use the measured Kd values for NMN and NaNM and the cellular concentrations of NMN and NaNM to calculate the occupancy of K61Q PncC with each metabolite. Basically, calculate the percentages of K61Q PnCC that would exist as K61Q PnCC-NMN complex and the K61Q PnCC-NaNM complex. If the calculations reveal a high percentage of the K61Q PncC-NMN complex, then the authors could argue the value of their work. With a Kd value of 100-200 µM, this reviewer suspects that percentage K61Q PnCC-NaNM complex will be low, particular if the cellular concentrations of NMN are in the 100 µM range.

Point 1: Without this analysis, this work seems more of a progress report than a successful study.

Response 1: We thank the referee for the suggestions.

In the revised version of the manuscript we have quantified the binding affinities of the mutant proteins from the thermal denaturation curves by using the approach suggested by Referee 1. These analyses confirmed that the both mutant proteins bind NMN in the micromolar range. They also revealed that K61Q mutant protein does not significantly interact with NaMN. In fact, NaMN does not significantly affect the mutant protein denaturation curve (Fig. 4a). It was not possible to calculate any Kd value. This is in agreement with the results obtained with the Trp fluorescence assay (Fig. 4d) and it confirms the scarce affinity of this mutant protein towards NaMN, and, consequently, its suitability as an MRE for an NMN sensor in a cellular setting.

Based on the available information on cellular NMN levels and the Kd values, we calculated the percentage of the protein that would be occupied by NMN under physiological concentrations. We found that the occupancy of the mutant with NMN would range from 4 % to 20 %. Therefore, K61Q would respond in a proportional manner to changes in cellular NMN levels.

Both the binding affinity and the specificity of K61Q protein for NMN confirm the usefulness of the biosensor in a cellular setting. The analysis of the mutant occupancy has been described in the Discussion section (lines 325-329) and in Materials and Methods section

Round 2

Reviewer 1 Report

Authors have taken into account all suggestions.